# Evaluation of Adjuvant Antibiotic Loaded Injectable Bio-Composite Material in Diabetic Foot Osteomyelitis and Charcot Foot Reconstruction

**DOI:** 10.3390/jcm12093239

**Published:** 2023-05-01

**Authors:** Venu Kavarthapu, Jasdeep Giddie, Varun Kommalapati, Joanne Casey, Maureen Bates, Prashanth Vas

**Affiliations:** 1Department of Orthopedic Surgery, King’s College NHS Foundation Trust, London SE5 9RS, UK; 2Diabetes Foot Clinic, King’s College NHS Foundation Trust, London SE5 9RS, UK

**Keywords:** diabetic foot ulcers, diabetic foot infection, Charcot foot reconstruction, diabetic foot osteomyelitis, biocomposite, calcium sulphate

## Abstract

The management of diabetic foot osteomyelitis (DFO) is extremely challenging with high amputation rates reported alongside a five-year mortality risk of more than fifty percent. We describe our experience in using adjuvant antibiotic-loaded bio-composite material (Cerament) in the surgical management of DFO and infected Charcot foot reconstruction. We undertook a retrospective evaluation of 53 consecutive patients (54 feet) who underwent Gentamicin or Vancomycin-loaded Cerament application during surgery. The feet were categorised into two groups: Group 1, with infected ulcer and DFO, managed with radical debridement only (*n* = 17), and Group 2, requiring reconstruction surgery for infected and deformed Charcot foot. Group 2 was further subdivided into 2a, with feet previously cleared of infection and undergoing a single-stage reconstruction (*n* = 19), and 2b, with feet having an active infection managed with a two-stage reconstruction (*n* = 18). The mean age was 56 years (27–83) and 59% (31/53) were males. The mean BMI was 30.2 kg/m^2^ (20.8–45.5). Foot ulcers were present in 69% (37/54) feet. At a mean follow-up of 30 months (12–98), there were two patients lost to follow up and the mortality rate was 11% (*n* = 5). The mean duration of post-operative systemic antibiotic administration was 20 days (4–42). Thirteen out of fifteen feet (87%) in group 1 achieved complete eradication of infection. There was a 100% primary ulcer resolution, 100% limb salvage and 76% bony union rate within Group 2. However, five patients, all in group 2, required reoperations due to problems with bone union. The use of antibiotic-loaded Cerament resulted in a high proportion of patients achieving infection clearance, functional limb salvage and decrease in the duration of postoperative antibiotic therapy. Larger, preferably randomised, studies are required to further validate these observations.

## 1. Introduction

Diabetic foot ulcers (DFU) are difficult to manage and can lead to major lower limb amputation (MLEA) with a mortality rate of over 50% at 5 years [1]. As a result, the surgical management of the diabetic foot has seen a resurgence of interest, not only for infection control surgery, but also for addressing structural foot deformities, with the goal of achieving a functional limb and reducing the risk of recurrent DFU [2,3]. However, such surgical intervention presents several significant challenges, including a compromised soft tissue envelope, delayed healing due to loss of protective sensation, impaired tissue oxygenation, and the likelihood of infection recurrence. The presence of diabetic foot osteomyelitis (DFO), whether accompanied by Charcot foot deformity or not, can pose a significant challenge to both clinicians and patients. This complexity often leads to suboptimal clearance of infection, which can result in the development of multi-resistant organisms that require prolonged and repeated antibiotic therapies [4]. As a result, effectively treating DFO requires a comprehensive and targeted approach.

The surgical debridement of an infected diabetic foot involves the removal of all visibly infected and necrotic tissues, including bone, resulting in a significant reduction in the local infection burden. If followed by targeted antibiotic therapy, complete eradication of any residual infection can be achieved. However, it is crucial to correct any coexistent significant deformity following infection eradication to prevent ulcer recurrence and maintain ambulatory status. Unfortunately, large bone resections during debridement of osteomyelitic bones and deformity corrective osteotomies may leave bone voids that can become a nidus for infection secondary to contiguous bacterial seeding from adjacent uncleared infected areas. This is particularly concerning as the penetration of systemically administered antibiotics into osseous voids has been shown to be poor and associated with suboptimal local drug concentrations. Equally, Charcot foot deformity correction has been demonstrated to significantly improve functional and quality of life outcomes [3,5]. However, addressing the bone voids that are often encountered during one-stage or two-stage approaches can have a notable impact on the success of the surgical outcome, including bone fusion and the durability of the correction achieved. Thus, effective management of these bone defects is critical for achieving optimal results in the treatment of DFO Charcot foot deformity correction.

The use of adjuvant antibiotic-loaded biodegradable vehicles to fill bone voids has the potential to address concerns regarding local antibiotic elution and bone formation stimulation [6]. Early experiences with a new antibiotic-loaded injectable biocomposite material (Cerament^®^, Bonesupport, Lund, Sweden) consisting of 60% Calcium Sulphate and 40% Calcium Hydroxyapatite have been encouraging in the treatment of chronic bone and joint infections [7]. However, the potential scope of this material is much wider [6], and its deployment in orthopaedic infection clearance is being increasingly explored [8,9], but its adoption in the surgical reconstruction of a previously osteomyelitic Charcot foot has not been reported, making it of great interest to those actively managing such patients. Therefore, we evaluated the effectiveness of antibiotic-loaded Cerament in eradicating infections and promoting bone healing during the surgical management of infected diabetic foot and in Charcot foot reconstructions.

## 2. Materials and Methods

### 2.1. Design

This was a retrospective service evaluation which collected information from case notes on patient demographics and co-morbidities, infection status, clinical features, investigations, surgical treatment, antibiotic treatment and the outcomes.

### 2.2. Inclusion Criteria

Consecutive patients under the care of the diabetes foot unit at King’s College Hospital that had Cerament application during surgery, between September 2015 and June 2019, were included in this study. Those with active DFU had University of Texas Wound Classification Grade 3 ulceration with evidence of DFO. As a minimum, the presence of DFO was suspected both through clinical and radiological examination and confirmed through positive microbiological growth.

### 2.3. Patient Groups

Foot presentations were divided into two main groups dependent upon the procedures undertaken: Group 1—feet with infected ulcer and DFO, managed with radical debridement; and Group 2—those that had reconstruction surgery for deformed and infected Charcot foot. The latter was subdivided into Group 2a—feet that had previous surgical clearance of DFO, undergoing a single stage reconstruction, and Group 2b—presented with actively infected deformed Charcot foot, managed with a staged reconstruction procedure. All individuals had application of antibiotic impregnated Cerament with either gentamicin (Cerament G) or vancomycin (Cerament V) during the surgical procedure.

Patients were managed by the multi-disciplinary diabetes foot team (MDFT), which included a diabetologist, orthopaedic surgeon, vascular surgeon, plastic surgeon, microbiologist and an orthotist as its core members.

### 2.4. Surgical Management

All surgical procedures were carried out by a senior orthopaedic surgeon from the MDFT, and in cases where feasible, pre-operative administration of antibiotics was delayed until multiple deep tissue samples were collected intraoperatively from all patients [10]. Aggressive debridement was performed in both groups, and all infected tissues were excised until the healthy bleeding margins of the green zone were reached [11].

All patients within group 1 were managed as a single stage procedure that included ulcer debridement and any additional procedures such as exostectomy.

Single stage reconstruction (group 2a) was chosen among the group of patients that had a previous history of infected ulcers and DFO and shown evidence of infection clearance following previous ulcer debridement, exostectomy, and administration culture specific targeted antibiotics [12,13]. During the reconstruction procedure, the Charcot deformity correction was achieved through bone osteotomies and wedge resections and the correction was maintained with internal fixation devices. All measures were taken to achieve optimal bone opposition during fixation through compression of the bone fragments [14,15]. However, it is recognised that there are often small bone voids and gaps between the bone fragments even after compressive fixation.

Group 2b staged procedures were reserved for patients with active deep tissue infections that required a formal surgical debridement as the first stage, prior to the definitive reconstruction procedure. Our unit has described the surgical technique and protocol in a case series [16]. In the first stage, after excision of osteomyelitic bone, osteotomies are performed to improve deformity, and temporary stabilisation of the osteotomies is achieved using threaded 2.5 mm to 3.5 mm guidewires or an external fixator. To provide a high concentration of antibiotic in the surrounding tissues, a local antibiotic-eluting calcium sulphate preparation (Stimulan, Biocomposites, Keele, UK) is used to fill the bone voids. We prefer this product over Cerament at the first stage as it is less expensive and the goal is to achieve antibiotic elution only, without the need for bone healing promotion. Wounds are left open as needed and managed with negative pressure wound therapy (NWPT). Targeted intravenous antibiotics are continued along with advanced wound care and offloading of the foot in a total contact cast (TCC).

When there was clinical and serological evidence of infection eradication, usually at 6–8-week mark, the second stage of reconstruction was performed. During the second stage of reconstruction, further bone resections were performed to accomplish optimal deformity correction, and skeletal stabilisation was achieved using appropriate internal fixation techniques [16].

### 2.5. Cerament Instillation

Prior to the wound closure, in group 1, 2a and 2b during the second stage, Cerament V or G was applied to the bone debridement and osteotomy areas. This was performed by injecting directly into any osseous voids created or within the medullary cavities of the metatarsals following drilling and curettage, or into multiple drill holes created within the bones in the infected areas, or a combination of these [8,17] (Figure 1a,b). Care was taken to create a dry bone bed while injecting Cerament to promote interdigitation once it was set. The choice of antibiotic was based on the sensitivity results of deep tissue specimen cultures (group 1 and group 2a) or previous intraoperative bone sampling cultures in group 2b (precedence given when available). If no clear pathogen was identified or previous microbiological results were inconsistent, a discussion with microbiology was undertaken to decide on the best antibiotic option.

The surgical wounds were primarily closed, and the ulcers were left open when closure was not possible. The open ulcers were managed with NWPT in a bivalved TCC initially, followed by a closed TCC when the NWPT was no longer required. The intravenous antibiotics were continued until there was clinical and serological evidence of infection clearance. Patients were initially followed up, following discharge, at two weekly intervals for regular change in a TCC, and at 6 weeks, 3 months, 6 months, 12 months and annually thereafter, with radiographs for the assessment of bony union. The bone healing was considered satisfactory when consolidation of three or more cortices or bone bridging across the fusion site of more than 50% was noted on plain radiographs taken in two orthogonal views. Computerised tomography scans were obtained if their roentgenograms showed signs of delayed or non-union or motion noted clinically at the fusion site (Figure 2a–e). Patients maintained non-weight bearing in a TCC until there was evidence of bone healing and then transitioned into bespoke surgical footwear. None of the patients in Group 2b were advised to wear commercial footwear with or without bespoke insoles.

### 2.6. Outcomes

Primary outcomes evaluated were the proportion of participants who achieved infection eradication and primary bone union. Secondary outcomes included ulcer resolution and recurrence, ambulatory status, limb salvage, mortality rate, and the need for orthopaedic re-intervention in the same area.

### 2.7. Governance and Approval

In discussion with the hospital trust Research & Innovation Department, and using the NHS Health Research Authority’s online tool, the project was deemed to be a service evaluation. Ethical approval was not required as patients’ management was not affected in any way and treatment had already been provided. All patients attending the King’s Diabetic Foot Service are consented to participate in research and audit projects, which include service evaluation.

### 2.8. Statistical Analysis

Statistical analysis was performed using IBM SPSS statistics version 24. Categorical variables were analysed using Fisher’s exact test and continuous variables were analysed using an independent samples *t*-test. Differences between the three groups for the variables described in the tables were determined using a one-way ANOVA. Statistical significance was set at *p* < 0.05.

## 3. Results

We identified 53 consecutive patients who underwent a surgical procedure using Cerament in a total of 54 feet. One patient underwent staged bilateral procedures. The mean age was 56 years (27–83) and 59% (31/53) were males. The mean BMI was 30.2 kg/m^2^ (20.8–45.5) including five patients with a BMI ≥ 40. All patients presented with peripheral neuropathy; in 72% (38/53) due to type 2 diabetes, 25% (13/53) due to type 1 diabetes and 2 patients (4%) due to Charcot Marie Tooth disease. In terms of diabetic complications, 53% (27/51) had retinopathy and 41% (21/51) chronic kidney disease stage 3 or higher including 4 on renal dialysis. Additionally, 14% (7/51) had a previous revascularisation procedure for peripheral arterial disease. Chronic DFU were present in 70% (37/53) patients. Prior to a review in our clinic, 47% (25/53) were recommended a major limb amputation at their local units. Patients were followed up in the multidisciplinary foot service for a mean duration of 30 months (range 12–98), with two patients lost to follow up, one from each group.

### 3.1. Procedures

Group 1 consisted of 17 feet (31% of total cohort). These included six patients for minor amputations (four transmetatarsal and two fifth ray amputations), six forefoot ulcer and bone debridement, four os calcis ulcer and bone debridement, and one hindfoot ulcer debridement, along with Achilles’ tendon lengthening and partial talectomy. Group 2 consistent of 37 feet of which 19 had a single stage reconstruction (group 2a, 35% of total cohort) whilst 18 required a staged reconstruction (group 2b, 33% of total cohort). There were no major differences in patient demographics between the three groups, as shown in Table 1.

### 3.2. Microbiology

The intra-operative bone and deep tissue specimens were analysed (Figure 3). *Staphylococci* sp. were the most common organism (30%). Gram-negative organisms were identified in 34% and a polymicrobial infection was seen in 25% of isolates with a combination of gram positive, negative and anaerobes. There was no growth in 15 samples (20%). Post-operative systemic antibiotics were administered for a mean 20 days (range 4–42).

### 3.3. Cerament Use

Cerament V was used in 39% (21/54) feet while 65% (35/54) feet had Cerament G instilled during surgery to cover the most significant isolates; the difference was statistically significant, *p* = 0.037. In two patients, both Cerament V and G were used.

## 4. Primary Outcome Measures

Two patients from group 1 were not included in the analysis as they died within 12 months from their surgery. An overview of post-operative results is provided in Table 2.

### 4.1. Infection Resolution and Ulcer Healing

In group 1, 13/15 (87%) feet achieved complete eradication of infection and ulcer resolution following the first surgical procedure. Two feet within this group had persisting ulcers and both patients elected to continue managing their ulcers non-operatively.

Within Group 2, all the primary ulcers resolved. There were no cases of ulcer persistence or reoccurrence at the index region. Five patients (two in group 2a and three in group 2b) within the reconstruction group developed de novo ulcers in other areas of the foot. Three of these patients had wounds attributed to metalwork prominences and underwent partial metalwork removal that resulted in wound healing. The remaining two patients developed forefoot ulcers that were treated with minor amputations.

### 4.2. Primary Bone Union

This was relevant for Group 2 only, and the overall bone fusion rate was 76%. In group 2a the primary bone union rate was 68% (*n* = 13/19), whereas it was 83% (*n* = 15/18) within group 2b. Details of patients not achieving primary union are shown in Table 3.

### 4.3. Post-Operative Infection Recurrence

One patient from group 2b developed a deep infection nine months following a staged reconstruction. This required removal of all internal metalwork and further radical debridement followed by reconstruction.

## 5. Secondary Outcome Measures

### 5.1. Metalwork Infection

There was one case of infected metalwork and bone non-union in group 2b, requiring metal work removal and further debridement.

### 5.2. Ulcer Recurrence

There were no cases of ulcer reoccurrence at the index site in any patient groups at the end of the follow-up. 

### 5.3. Post-Operative Ambulation

Thirty-six patients (36/51, 71%) achieved full weight bearing with orthotic shoes. A total of twelve patients (24%) required custom made ankle foot orthosis, seven of which were full weight bearing and five patients were partial weight bearing. Three (6%) patients remained wheelchair bound.

### 5.4. Mortality

There was a total of five deaths during the follow-up period (mortality rate 11%), two of which were in group 1, and occurred within 12 months of their surgery. None of the deaths were related to the surgery or occurred within the first 3 months post-operatively.

### 5.5. Limb Salvage

Within this case series, we report a 100% limb salvage rate at the end of the follow-up period (mean duration of 30 months). No major amputations were undertaken in any of those who died.

### 5.6. Further Procedures

Of the five cases with de novo ulcers that were treated with metalwork removal (*n* = 3) and minor amputations (*n* = 2), there were a further five operations performed due to bone union issues (Table 3). One patient had a delayed union at the ankle following a hindfoot nail fixation (group 2a) and was successfully treated through dynamisation of the nail. Three patients had aseptic non-unions (two hindfoot and one midfoot), of which two had broken hindfoot nails, requiring removal and revision hindfoot fusions, and one midfoot non-union required an exostectomy. One patient had an infected non-union (group 2b) that was treated with removal of metal work and further debridement.

## 6. Discussion

Prevention of infection recurrence following the surgical management of DFO remains a challenge. Complex diabetic foot infection clearance procedures often require aggressive bone resections and osteotomies, and the management of resultant dead space is considered critical. Local antibiotics delivery into these bone voids is a developing area in the management of diabetic foot infections, offering evident benefits. While non-biodegradable products such as polymethylmethacrylate (PMMA) has been utilised in the past, more recently, the easy availability of calcium sulphate derived products (natural or composite), which are biodegradable, serve as the quintessential platform, providing greater apparent safety and excellent drug release kinetics when impregnated with a range of antibiotics [6,18]. Therefore, filling of the dead space with local antibiotic eluting synthetic biodegradable bone substitutes, may help achieve infection eradication. It has been identified that this method delivers high concentrations of antibiotics locally delivered, often in the order of 10–100 times the minimum inhibitory concentration (MIC) [19,20] and potentially above the mean antimicrobial eradication concentration, without systemic toxicity. Some diabetic foot infection clearance procedures, such as Charcot deformity corrections, benefit from additional bone healing stimulation that can promote healing of osteotomies. Bone grafts that are typically used for such a purpose carry a high risk of contracting infection from bacterial seeding from adjacent previously infected areas and are generally not recommended in the presence of previous infections [6,18]. Injectable calcium hydroxy apatite material can lead to osseoconduction and promote bone healing. Cerament is an injectable and completely resorbable Calcium sulphate and hydroxyapatite biocomposite that can have added Gentamycin or Vancomycin, thus providing both local antibiotic elution and bone stimulation. Cerament impregnated with antibiotics has been shown to be effective in chronic osteomyelitis [7], but its potential in the surgical treatment of complex diabetic foot osteomyelitis has only recently been investigated [8,9,21,22]. However, the use of Cerament in Charcot reconstruction for its additional positive effect on bone healing has yet to be explored.

An important highlight Is that our report represents the first series describing the use of Cerament in diabetic foot Charcot reconstruction surgery. The rate of bone fusion in our current study was 76%, limb salvage was 100% and independent ambulation was 90%. Comparison with previous published literature is difficult, as a large proportion of our cohort (43%) had simultaneous Charcot midfoot and hindfoot reconstruction, a significantly greater undertaking than the previous series which reported a fusion rate of 90% with 100% limb salvage involved *hindfoot only* [12]. However, simultaneous mid and hindfoot arthrodesis has been shown to have a 12 times higher rate of non-union and metal work breakage, compared to isolated hindfoot or midfoot [23]. Five out of eight patients (63%) in the non-union group developed a stable pseudoarthrosis with a deformity free plantigrade foot and no further treatment was required. Cerament also contain hydroxyapatite which is recognised for its ability to support bone formation and act as an osteoconductive scaffold. According to Nilsson et al., [24] in vitro biomechanical tests indicate that Cerament G has compression strength comparable to cancellous bone and promotes bone growth. However, despite the use of Cerament, our study’s group experienced a 24% non-union rate, raising the possibility that the healing response to use of biocomposite bone substitute may be suboptimal in infected Charcot foot bones compared to those that are healthy and normal. We believe our results provide a foundation for further research exploring the efficacy of Cerament as an osteoconductive scaffold in the surgical reconstruction of Charcot foot.

Achieving durable infection eradication typically requires surgical debridement of infected, non-viable bone and soft tissues. However, current guidelines recommend [25,26], and, indeed, specialist centres provide, concurrent systemic antibiotic therapy to ensure the eradication of any remaining infection. The optimal duration for such therapy is uncertain [4,26], and patients are frequently offered extended antibiotic regimens. The mean duration of systemic antibiotic administration in our study was 20 days (range 4–42), which is considerably lower than other reported studies [7,17]. Earlier series from McNally et al. [7] and Drampalos et al. [17] received post-operative systemic antibiotic administration for between 6 to 12 weeks after the indexed procedure. Similarly, in two studies where a ring fixator was used to correct Charcot deformity and achieve stability, the postoperative systemic antibiotic therapy was for 8 and 11 weeks, respectively [27,28]. Two recent studies using Cerament in DFO have reported using systemic antibiotics between 4 and 6 weeks, indicating a possible trend that clinicians are now becoming more confident limiting systemic antibiotic duration with antibiotic impregnated Cerament instillation. Our findings taken together with the previously published reports, make a stand for antibiotic stewardship and further support the results from two recent randomised controlled trials on diabetic foot infections [29,30], challenging the notion prevalent within DFO care that long duration of systemic antibiotics are ostensibly required, if durable infection clearance is to be achieved.

Wound ooze and inflammatory reactions with calcium sulphate-based void fillers has been well described in the literature [31]. McNally reported a wound leak of 6% in his series of 100 patients [7]. We have seen a decreasing trend in both wound ooze and ‘Cerament burns’ (skin erythema from leakage of Cerament) amongst our patients, which we attribute to the learning curve associated with its use. We meticulously place the Cerament within the osteotomy site prior to the application rigid compression or intraosseously via drill holes, limiting the amount of Cerament leakage within the soft tissue envelope.

The use of Cerament was distinctly different within our groups. In Group 1, the main aim was infection eradication and wound closure at the time of surgery. In Group 2a, infection control had already been achieved through previous surgery and the patients were subsequently subjected to deformity correction procedure, during which Cerament was used to eradicate any residual infection and promote bone healing to achieve bone union. Similarly, Cerament was used among group 2b patients during the second stage of the procedure, with the aim of achieving eradication of any residual infection and promoting bone fusion. Cerament was not used during the first stage of two stage as it is more expensive and there was no requirement for bone fusion during this state.

The antibiotic admixed with Cerament did not unequivocally match the intra-operative microbiological isolates. The decision on the best suited antibiotic for intra-operative instillation was based on preceding microbiological data, which have shown to have only fair to moderate concordance with surgical bone specimens [32]. Furthermore, polymicrobial growths, commonly prevalent in DFO, can often limit determination of the most important isolate to target. This underscores the importance of rigorous perioperative planning, including detailed discussions with microbiology colleagues to determine the appropriate choice of antibiotic to admix with Cerament. In addition, we continued systemic antibiotics targeted against isolates from surgical specimens, until inflammatory markers were deemed controlled but not normalised.

A surprising and noteworthy finding was that the mortality rate among our cohort was lower than expected. All our patients were referred to us after a period of uncontrolled infection and DFO development and typically such severe presentations are linked to high mortality rates. For instance, the United Kingdom National Diabetic Foot audit recorded a 14% mortality rate after 12 months among individuals with severe ulceration at presentation [33,34]. In a recent study, we reported a mortality rate of up to 45% after 18 months among a cohort that experienced a diabetic foot attack requiring urgent surgical debridement [35]. In contrast, the patients in our current study were younger, did not have advanced peripheral artery disease (PAD), and had chronic low-level infections instead of severe infections compared to the other groups. However, our evaluation was not designed to investigate the impact on mortality and, along with the small sample size, these factors could have contributed to the lower mortality rate. Another possible explanation could be that ensuring complete infection resolution and maintaining mobility through deformity correction may have contributed to extended survival by reducing chronic inflammation.

The strength of our study includes the reporting of outcomes from a diabetic foot centre with a well-established pre-and post-operative protocols in the surgical management of diabetic foot infections. We have analysed the clinical outcomes, including infection eradication and complications, functional outcome, including ambulatory status and radiological outcomes on this group of complex presentations. Limitations include the retrospective nature of our evaluation, the relatively low numbers of patients within each group and that we did not have a comparator group. Nonetheless, at present, this represents the largest reported series on the use of Cerament in diabetic foot reconstructive surgery.

In conclusion, we report on our experience utilising Cerament in DFO surgery and also, for the first time, in diabetic foot Charcot reconstructive surgery. The use of Cerament resulted in high proportion of functional limb salvage and infection clearance and decrease in the duration of post-operative antibiotic therapy. In addition to improvement in functional status, we observed apparent improved survival of the individual with diabetic foot disease. Further studies, ideally larger controlled cohorts, carefully exploring these observations with Cerament instillation in complex diabetic foot orthopaedic interventions are required.

## Figures and Tables

**Figure 1 jcm-12-03239-f001:**
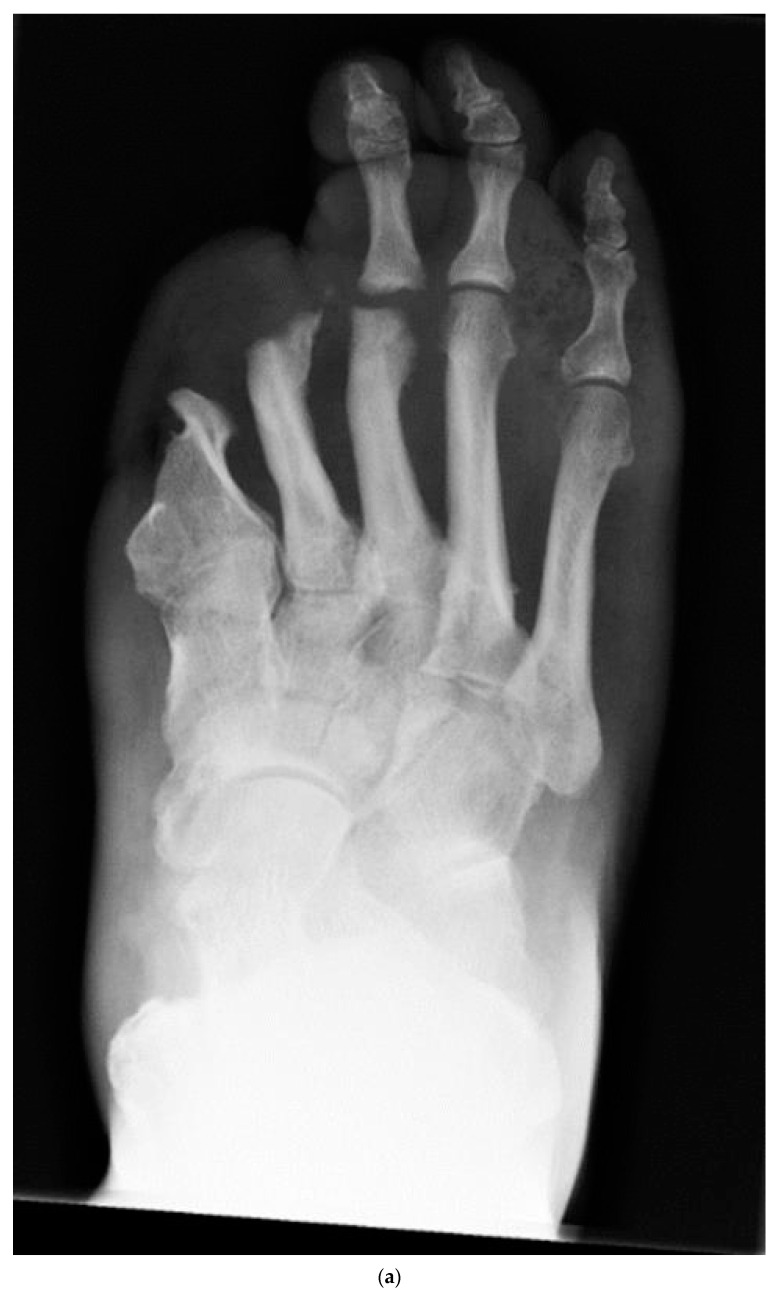
(**a**,**b**): Dorsoplantar radiographs of the foot taken before surgery (**a**) demonstrating osteomyelitic changes in the forefoot, and following trans-metatarsal amputation demonstrating Cerament application in the metatarsal stumps.

**Figure 2 jcm-12-03239-f002:**
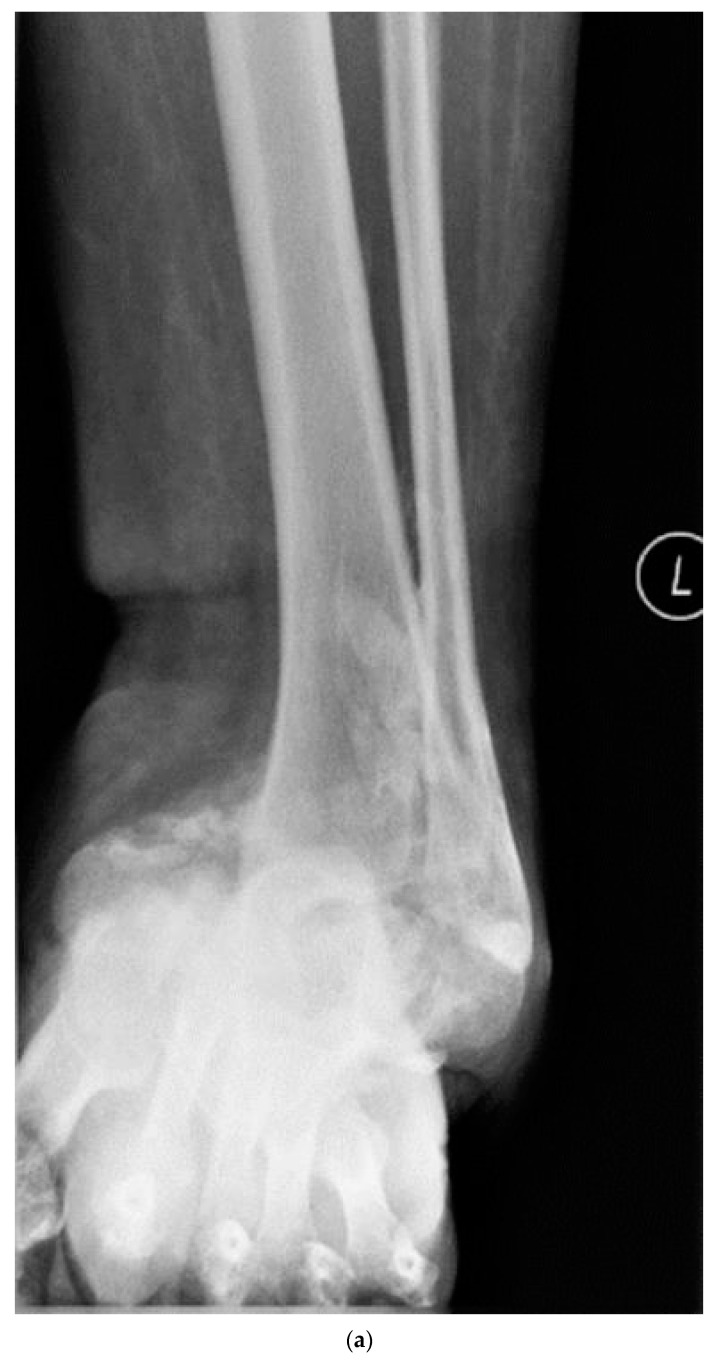
(**a**–**e**): Pre-operative weight bearing AP (**a**) and lateral (**b**) radiographs showing Charcot hindfoot and midfoot. Intra-operative fluoroscopy image (**c**) showing application of Cerament (arrow) following fixation. Foot and ankle lateral radiograph (**d**) and foot oblique radiograph taken at 6 months following reconstruction demonstrating bony union.

**Figure 3 jcm-12-03239-f003:**
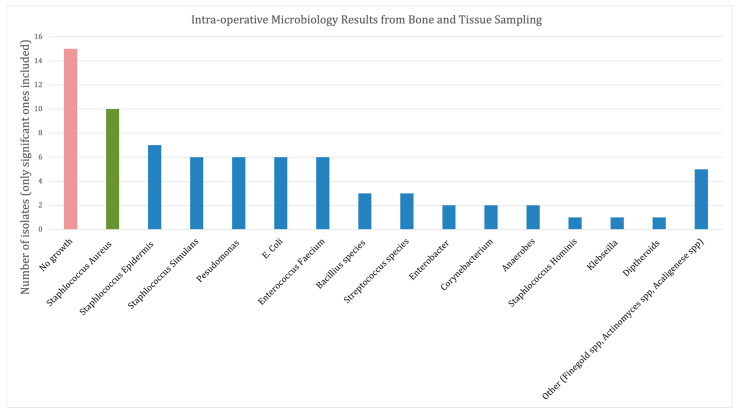
Intra-operative microbiology results from bone and tissue sampling.

**Table 1 jcm-12-03239-t001:** Pre-operative Patient Details. BMI = body mass index, ASA = American Society of Anaesthesiology score, eGFR = estimated glomerular filtration rate, NS = non-significant.

	Group 1	Group 2a	Group 2b	One-Way ANOVA
Number of patients (*n*=)	17	19	17	NS
Number of feet operated upon (*n*=)	17	19	18	NS
Number of patients lost to follow up	0	1	0	NS
Number of Males (*n*=)	12	11	8	NS
Mean age (years)	55.7	55.7	55.8	NS
Mean BMI (kg/m^2^)	28.9	33.3	31.7	NS
Number of pre–operative ulcers (*n*=)	16	8	13	NS
Mean ASA	2.4	2.5	2.6	NS
Retinopathy	9	9	9	NS
Nephropathy (eGFR <30 mL/min)	4	7	6	NS
Renal Dialysis	0	2	2	NS
Preceding revascularisation (*n*=)	0	3	4	NS
**Pre-operative Mobility:**				
Independent	8	3	1	*p* < 0.05
Stick	3	6	4	NS
Wheelchair	6	10	12	NS

NS = non significant.

**Table 2 jcm-12-03239-t002:** Post-operative Results. NA = not applicable, NS = non-significant.

	Group 1A	Group 2a	Group 2b	One-Way ANOVA
Primary Bone Union	NA	13	15	NS
Non-Unions	NA	6	3	NS
Post-operative Deep Infection	None	None	1	NS
Non healing ulcer	2	None	None	NS
New Ulcer Formation (not at index site)	None	2	3	NS
Mortality within each group	2	1	3	NS
Mean post-operative Haemoglobin	109	98	103	NS
Post-operative ambulatory status:				NS
Independent in orthotic shoes	10	13	13	NS
Independent in a bivalve cast	1	3	3	NS
Partial weight bearing	3	1	1	NS
Non weight bearing/wheelchair	1	1	1	NS

**Table 3 jcm-12-03239-t003:** Details of Non-Unions.

Case	Single/Two Stage	Reconstruction Location	Smoking Status	Clinical Stability	Further Procedures
1	Single	Midfoot	Ex-Smoker	Stable	Removal of metalwork—bolt to prevent further ulceration
2	Single	Mid and hindfoot	No	Unstable	Exostectomy
3	Single	Midfoot	Ex-smoker	Stable	None
4	Single	Mid and hindfoot	No	Stable	None
5	Single	Hindfoot	Yes	Unstable	Removal of broken nail and revision hindfoot fusion
6	Single	Midfoot	No	Stable	None
7	Two stage	Mid and hindfoot	No	Unstable	Revision of hindfoot nail and ulcer debridement
8	Two stage	Mid and hindfoot	Yes	Stable	None
9	Two stage	Midfoot	Yes	Infected non union	Removal of metalwork and repeat stage 1 procedure due to deep infection

## Data Availability

No data is provided due to privacy restrictions.

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
