# Peer review of "Evaluation of Adjuvant Antibiotic Loaded Injectable Bio-Composite Material in Diabetic Foot Osteomyelitis and Charcot Foot Reconstruction"

_jcm, 2023, doi:10.3390/jcm12093239_

Round 1

Reviewer 1 Report

The topic of the study is interesting; DFO is a major complication in orthopedic clinical practice and has a major cost on the health care system. The research is based on 53 patients (54 feet) who were grafted with antibiotic-loaded Cerament. The methodological approach appears correct, the data are clear and well reported in the tables. The introduction could benefit from more discussion of the pathology, and the discussion could benefit from some citations of recent literature reviews of Charcot disease and its treatment:

PMID: 27587513

PMID: 36769345

PMID: 32913602

It is also recommended that the bibliography, which does not appear clear, be improved.

Author Response

Thank you for the review. We agree with the suggestions and have made adjustments throughout the manuscript in line with the suggestion, although to maintain focus on Cerament as the key driver of outcomes, we have kept the Charcot pathology discussion limited. We hope the reviewer is agreeable. Additionally, we have fully redone the references in line with MDPI requirements.

Reviewer 2 Report

This is a high-quality research and the topic is quite suitable for this journal, which deals with a retrospective research related to a new biomaterial applied in the treatment for diabetic foot. I have only one concern:

I believe it will be more comprehensive if the authors could add some new advance of different kinds of biomaterial application in the disease in the discussion.

Author Response

We thank the reviewer. While we have maintained focus on Cerament as the key driver of outcomes, a brief section on biodegradable and nonbiodegradable substitutes has now been included in the discussion section. We hope the reviewer is agreeable. Additionally, we have fully redone the references in line with MDPI requirements, which includes reviews exploring bone substitutes.

Reviewer 3 Report

Dear authors, thank you very much for the opportunity to review your novel and interesting research.

The current research describes one of the longest retrospective cohorts of Charcot foot patients surgically reconstructed and treated with antibiotic loaded biocomposite divided by different groups. 

The research was divided into different groups; 1) infected ulcers including diabetic foot osteomyelitis in the foot and, 2) infected ulcers including charcot foot with om in a single stage or two stage reconstruction procedures.

- The abstract is well designed and summarizes the main points of the research. Authors should check journal policies regarding abstracting and reduce word account up to 200 words. Additionally, remove headings from the text.

- The Introduction should refer previous literature regarding bone support with antbiotic loaded biocomposites, it may improve the quality of the rationale (Whisstock C, Volpe A, Ninkovic S, Marin M, Meloni M, Bruseghin M, Boschetti G, Brocco E. Multidisciplinary Approach for the Management and Treatment of Diabetic Foot Infections with a Resorbable, Gentamicin-Loaded Bone Graft Substitute. J Clin Med. 2020 Nov 6;9(11):3586.)

Main objective is well described.

Research design and methods is correctly structured, just a few questions that may improve the soundness of the paper:

- The authors state that  all individuals had application of atb impregnated cerament with either genta or vanco during the surgical procedure. How was the election of genta or vanco for patients treated in groups 1 and 2a? I understand that patients in group 2b were treated based on bone culture, but what about 1 and 2a?

- As secondary outcome measure authors analyzed ulcer recurrence, but how did the authors prescribed footwear after healing and Charcot foot stabilization? Did patients were prescribed with customized insoles? what about custom made footwear? Please add this topic to the methods.

Regarding results section:

- Authors declare that there do not exist any difference between groups regarding demographic characteristics, did authors performed any statistical test to ensure that affirmation? If no, please perform it consequently.

- Table 1 should be reformatted to mdpi characteristics, additionally add abbreviations.

- Please perform statistical comparative between groups in table 2.

- Ulcer recurrence was 0% for all the patients? How much was the median follow-up time for all the sample? I think this is an impressive result in a 30 month follow-up period. Congrats on it, please explain in methods how were the patients followed during the 30 month period? Did you follow IWGDF recommendations?

Discussion should be shortened and the retrospective basis of the research should be stated as a limitation.

Authors must fill properly author contribution. Did the research received any grant or funding? There exist any conflict of interest?

Additionally, did the research has institutional review board? Please clarify all lacking info.

I congratulate authors for the interesting and powerful data of your research. I recommend authors to improve the paper.

Author Response

Comments and Suggestions for Authors

Dear authors, thank you very much for the opportunity to review your novel and interesting research.

The current research describes one of the longest retrospective cohorts of Charcot foot patients surgically reconstructed and treated with antibiotic loaded biocomposite divided by different groups. 

The research was divided into different groups; 1) infected ulcers including diabetic foot osteomyelitis in the foot and, 2) infected ulcers including charcot foot with om in a single stage or two stage reconstruction procedures.

- The abstract is well designed and summarizes the main points of the research. Authors should check journal policies regarding abstracting and reduce word account up to 200 words. Additionally, remove headings from the text.

Our response:

We thank the reviewer for pointing this out. The abstract has been accordingly shortened.

- The Introduction should refer previous literature regarding bone support with antbiotic loaded biocomposites, it may improve the quality of the rationale (Whisstock C, Volpe A, Ninkovic S, Marin M, Meloni M, Bruseghin M, Boschetti G, Brocco E. Multidisciplinary Approach for the Management and Treatment of Diabetic Foot Infections with a Resorbable, Gentamicin-Loaded Bone Graft Substitute. J Clin Med. 2020 Nov 6;9(11):3586.)

Our response:

We thank the reviewer for their suggestion. We have added it in the reference and also added it to the discussion section. We believe the papers on Cerament, despite being Level IV evidence, are pointing to a trend of superior antibiotic stewardship and reporting shorter systemic antibiotic regimes alongside more improved outcomes.

Main objective is well described.

Research design and methods is correctly structured, just a few questions that may improve the soundness of the paper:

- The authors state that  all individuals had application of atb impregnated cerament with either genta or vanco during the surgical procedure. How was the election of genta or vanco for patients treated in groups 1 and 2a? I understand that patients in group 2b were treated based on bone culture, but what about 1 and 2a?

Our response:

We thank the reviewer for their enquiry. We had previously very briefly discussed this in the discussion, but in the revised manuscript also included this in the methods. The reviewer will probably agree that this was (and is) not an exact science and we went with the best quality tissue sample analysis which the MDT felt was most indicative. Of course, for the 2b pts we had access to intraop specimen results. The choice of antibiotic to use will be a decision-making skill every expert will have to develop/learn on their own when they adopt Cerament or indeed any bone substitute.

- As secondary outcome measure authors analyzed ulcer recurrence, but how did the authors prescribed footwear after healing and Charcot foot stabilization? Did patients were prescribed with customized insoles? what about custom made footwear? Please add this topic to the methods.

Our response:

We thank the reviewer for their suggestion. We have added it.

Regarding results section:

- Authors declare that there do not exist any difference between groups regarding demographic characteristics, did authors performed any statistical test to ensure that affirmation? If no, please perform it consequently.

Our response:

We thank the reviewer for their suggestion.

- Table 1 should be reformatted to mdpi characteristics, additionally add abbreviations.

Our response:

We thank the reviewer for their suggestion. We have reformatted it.

- Please perform statistical comparative between groups in table 2.

Our response:

We thank the reviewer for their suggestion. We have included it but there are no differences in one-way ANOVA between the 3 groups.

- Ulcer recurrence was 0% for all the patients? How much was the median follow-up time for all the sample? I think this is an impressive result in a 30 month follow-up period. Congrats on it, please explain in methods how were the patients followed during the 30 month period? Did you follow IWGDF recommendations?

Our response:

We thank the reviewer for their enquiry. Yes, it was over 30 months. We have made it clear. As a unit, we always adhere to IWGDF guidance and indeed, aspire to be future benchmark setters.

Discussion should be shortened, and the retrospective basis of the research should be stated as a limitation.

Our response:

We thank the reviewer for their suggestion. We have improved on the discussion extensively. However, the other two reviewers request additional information which meant that the discussion section remained the same size. I hope the reviewers are agreeable to this and will enjoy reading our discussion as much as we did writing it. The limitation has been included.

Authors must fill properly author contribution. Did the research received any grant or funding? There exist any conflict of interest?

Our response:

The authors’ contribution has been declared in the manuscripts. All other relevant paperwork will be completed by all authors as needed. Our unit has received an unconditional grant from BoneSupport to support this study.

Additionally, did the research has institutional review board? Please clarify all lacking info.

Our response:

We thank the reviewer for their enquiry and we have added a separate section in the methods.

Round 2

Reviewer 3 Report

Dear authors, thank you very much for responding all the queries.